# Reliability of the thumb localizing test and its validity against quantitative measures with a robotic device in patients with hemiparetic stroke

Eri Otaka[1], Yohei Otaka[1,2]*, Shoko Kasuga[3,4], Atsuko Nishimoto[1], Kotaro Yamazaki[1], Michiyuki Kawakami[1], Junichi Ushiba[3,5], Meigen Liu[1]

1 Department of Rehabilitation Medicine, Keio University School of Medicine, Shinjuku-ku, Tokyo, Japan, 2 Department of Rehabilitation Medicine I, School of Medicine, Fujita Health University, Toyoake, Aichi, Japan, 3 Keio Institute of Pure and Applied Sciences (KiPAS), Keio University, Yokohama, Kanagawa, Japan, 4 Graduate School of Science and Technology, Keio University, Yokohama, Kanagawa, Japan, 5 Department of Biosciences and Informatics, Faculty of Science and Technology, Keio University, Yokohama, Kanagawa, Japan

* otaka119@mac.com

**Data Availability Statement:** All relevant data are within the manuscript and its Supporting Information files.

## Abstract

### Objectives

To examine the inter-rater reliability of the thumb localizing test (TLT) and its validity against quantitative measures of proprioception.

### Methods

The TLT was assessed by two raters in a standardized manner in 40 individuals with hemiparetic stroke. Inter-rater reliability was examined with weighted Kappa. For the quantitative measures, a bimanual matching task in a planar robotic device was performed. Without vision, each participant moved the unaffected hand to the perceived mirrored location of the affected hand, which was passively moved by the robot. Three measures were taken after 54 trials: *Variability*, trial-to-trial variability of the mirrored-matched locations; *Area*, the ratio of the area enclosed by the active hand relative to the passive hand; and *Shift*, systematic shifts between the passive and active hands. The correlation between the TLT and each robotic measure was examined with Spearman's rank correlation coefficient.

### Results

The overall weighted kappa of the TLT was 0.84 (*P*<0.001). The TLT correlated highly with *Area* (r = -0.71, *P*<0.001) and moderately with *Variability* (r = 0.40, *P* = 0.011). No significant correlation was found between the TLT and *Shift*.

### Conclusions

The TLT had a high inter-rater reliability, and was validated against quantitative measures of proprioception reflecting the perceived area of movement and variability of the limb location.

**Funding:** This study was supported in part by collaborative research funds from Toyota Tsusho Corporation (https://www.toyota-tsusho.com). The funder had no role in study design, data collection and analysis, decision to publish, or preparation of the manuscript.

**Competing interests:** This study was supported in part by collaborative research funds from Toyota Tsusho Corporation. M.L., M.K. and J.U. are the founding scientists of Connect Inc., a commercial company for the development of rehabilitation devices since May 2018. They have received a salary from Connect Inc., and have held shares in Connect Inc. They currently hold managerial positions at Connect Inc. These conditions were disclosed to the Universities, and were approved. Connect Inc. does not have any relationship with the present study. This does not alter our adherence to PLOS ONE policies on sharing data and materials.

## Introduction

Proprioception is necessary for the control of limb posture, and the coordination of multijoint movements [1–6]. In stroke patients, proprioceptive deficits are quite common [7, 8] and known to be partly related to motor recovery [9, 10]. As its longitudinal process of recovery has been partly revealed recently [8, 11, 12] and novel rehabilitative approaches are sought more than ever in clinical practice, indispensability for robust assessment tools of proprioception is increasing to accurately evaluate its improvement with progress of time or by any intervention.

Conventionally, the thumb localizing test (TLT) [13–15] or similar techniques [9, 16] are widely used for the assessment of proprioception for patients with stroke, both in clinical and research settings. The TLT is a test for "limb localization" and is clinically useful in that it can examine invisible perceptual deficits using a visible motor task and be easily performed at the patient's bedside, although it is a four-point ordinal scale and thus disadvantageous as a quantification tool in detecting subtle deficits or changes over time compared with continuous scales. In the development of the TLT, Hirayama et al. reported that the TLT showed greater frequency of abnormality than other physical examinations for proprioceptive sensation [15], and the TLT deficits had strong categorical relationship with joint position and movement test, and tactile cutaneous localization test [14]. However, reliability of the TLT has not been precisely shown. Furthermore, validity of the TLT has yet to be revealed because there has been no other conventional assessment of proprioception that is reliable and quantitative enough to be regarded as a reference.

Several groups have designed reliable and objective procedures to quantify proprioception [17–19]. Among these, the arm position matching task [20] provided by the KINARM Exoskeleton robotic device (Kinarm, Kingston, ON, Canada) [21] is a robotic assessment which describes kinematic characteristics of the individual's mirror-matched arm movement, visualizing and quantifying proprioception using a motor task same as the TLT. The measures derived from this robotic task have been shown to be adequately reliable [20] and correlated with the measures of activities of daily living and dexterity in patients with subacute stroke [22], enough to be considered as a reference in a validation study for other assessments. Regarding the TLT, only categorical relationships have been shown with some types of quantitative measures driven by the robotic device [20, 22] in the mixed group of healthy adults and stroke patients. However, no attempt has been made to examine the correlation between the TLT and quantitative measures driven from the device.

To confirm the robustness of the TLT as a clinical measure of proprioception, the present study aimed (1) to examine the inter-rater reliability of the TLT, and (2) to validate the TLT by examining correlation with the quantitative measures derived from the exoskeleton robotic device.

## Materials and methods

### Participants

The recruited participants included 40 patients with chronic stroke who visited the Department of Rehabilitation Medicine at Keio University Hospital for the rehabilitation of upper extremity paresis from July 2013 to January 2015. Patients with the following criteria were included: more than 150 days since stroke onset; stroke in single cerebral hemisphere; no clinically obvious cognitive deficits; living independently; absence of pain in the paretic upper limb; no impairments in the non-paretic upper limb; ability to maintain sitting position without difficulty; and adequate range of motion in the upper extremity to perform tasks in the

KINARM. Patients with history of epilepsy, implants (pacemakers, shunts, or clipping), and those who could not perform the KINARM tasks owing to pain and/or contractures were excluded.

The study protocol was approved by the institutional ethics committee (Keio University School of Medicine Ethics Committee #20120070). All participants gave written informed consent prior to their participation, according to the Declaration of Helsinki.

## Clinical assessments

To evaluate the participants' upper limb impairments, we assessed the affected arm using the Fugl-Meyer Assessment, the part A score of the upper limb section to quantify motor function (0 (worst) - 36 (best)) [23], and the Modified Ashworth Scale for elbow flexors to grade spasticity (0(no spasticity), 1, 1+, 2, 3, 4(rigid)) [24].

## Thumb localizing test

In accordance with Hirayama et al. [13, 14], we adhered to the following procedure: an examiner positioned the affected upper limb of the participant at a certain position, and asked participants to pinch the tip of the thumb with the unaffected fingers. For the standardization, three limb positions were defined (Fig 1), and all participants were assessed in the same order. Before each limb position was fixed, the examiner moved the limb multi-directionally to prevent the subject from guessing the limb position based on the starting position. The participants performed the procedure twice, with vision occluded condition (with a blindfold) and with vision restored condition (without a blindfold). This was to analyze the effect of vision, as the importance of visual information in compensating proprioceptive deficit is clinically well known [4, 25].

To score the TLT, all performances were recorded on videotape and were independently rated afterwards by two physiatrists; EO, with 6 years of clinical experience and YO, with 18 years of clinical experience. Three physiatrists, EO, YO, KY, with 5 years of clinical experience, and an occupational therapist, AN, with 12 years of clinical experience, served as examiners. As the descriptions of the TLT ratings are slightly different in two papers written by the original authors [13, 15], we adopted the rating of the more recent paper [13]: 0, normal (the reaching limb can go straight and quickly toward the tip of the fixed thumb and pinch it easily); 1, slightly impaired (the fingers are brought several centimeters apart from the fixed thumb and then corrected immediately, or the reaching limb sways on the way); 2, moderately impaired (the reaching limb seeks around the fixed thumb for more than several centimeters, touches the thumb by chance, or reaches the tip of the thumb by tracing from the fixed hand or other fingers); 3, severely impaired (cannot reach the fixed limb at all, or the reaching limb touches the fixed arm by chance and reaches the tip of the thumb by tracing the skin).

## Quantitative assessment of proprioception with the robotic device

**Robotic device.**   The KINARM Exoskeleton was used for robotic assessment (Fig 2A). This is a planar robotic device, which has one degree-of-freedom for the shoulder joint (horizontal adduction/abduction) and one for the elbow joint (flexion/extension). It allows free horizontal arm movement, and can monitor shoulder and elbow motion [21]. Participants were initially seated in a modified wheelchair base with each arm set in a trough so that the arms were approximately in the same plane as the shoulder (80–90˚ abducted), the examiner then slid the whole wheelchair into a planar monitor. Visual feedback or occlusion can be given on the virtual reality display just above the plane of the arms.

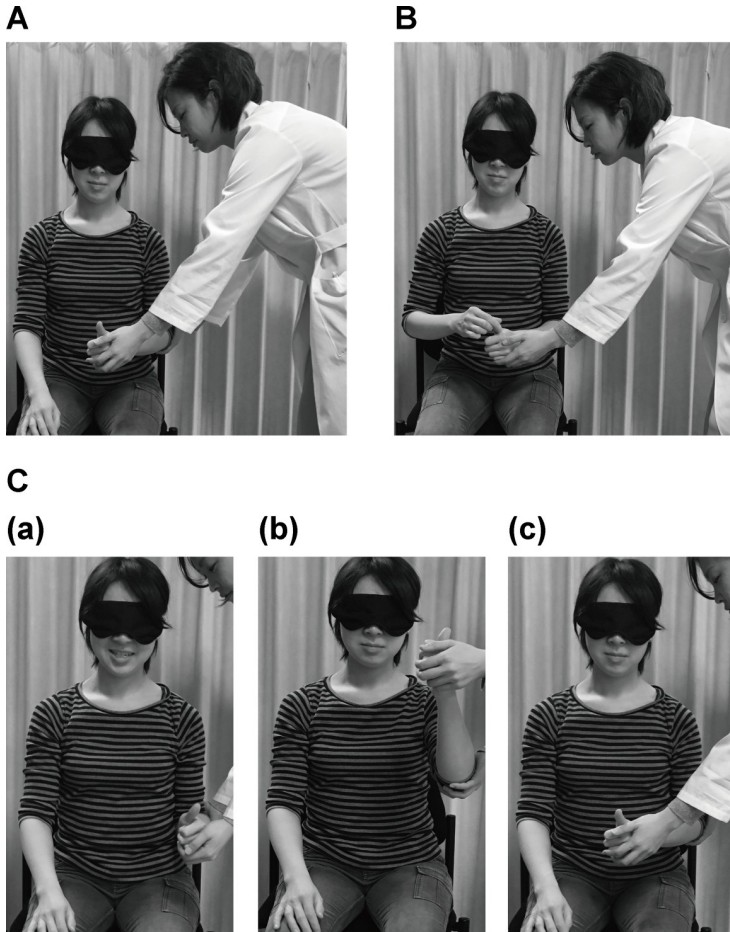

**Fig 1. Procedure of the thumb localizing test (TLT).** A. The reaching limb is on the ipsilateral knee at the start of the test. B. Participants are instructed to pinch the opposite thumb. C. The limb position of the fixed limb: (a) The forearm in the neutral position, with the elbow at 90˚ of flexion and the shoulder at 0˚ of flexion. (b) The forearm in the neutral position, with the elbow at 90˚ of flexion and the shoulder flexed so that the thumb is at the same level as the mouth. (c) The forearm in the neutral position, with the elbow at 90˚ of flexion and the shoulder internally rotated so that the thumb is over the midline of the trunk. The individuals pictured in this figure have provided written informed consent (as outlined in PLOS consent form) to publish their image alongside the manuscript.

**Assessment of proprioception.** The arm position matching task is a bimanual matching task [20]. The central target, which is the starting position, was set such that the shoulder was at approximately 30º of horizontal abduction and the elbow at 90º of flexion. The subject was instructed to relax the affected arm (passive limb) and allow the robot to move the arm in the horizontal plane to adopt a certain position. The subject was asked to then move the unaffected arm (active limb) to the perceived mirrored location. Starting from the preceding position, each participant performed the matching tasks consecutively for nine spatial targets separated by 10 cm for each block, and six such blocks for a session (total of 54 trials in a session); the order of the target positions in each block was pseudo-randomized. Similar to the TLT, all the participants completed the session twice: first with a shield above the arms so that vision was occluded, and then repeated the task with vision restored (Fig 2B). It took approximately 10–15 minutes per session, and all sessions were usually performed consecutively on the same day. Examiners routinely advised participants to take enough time and rest between the testing sessions to fully concentrate on the task, and most participants needed no rest or only a few

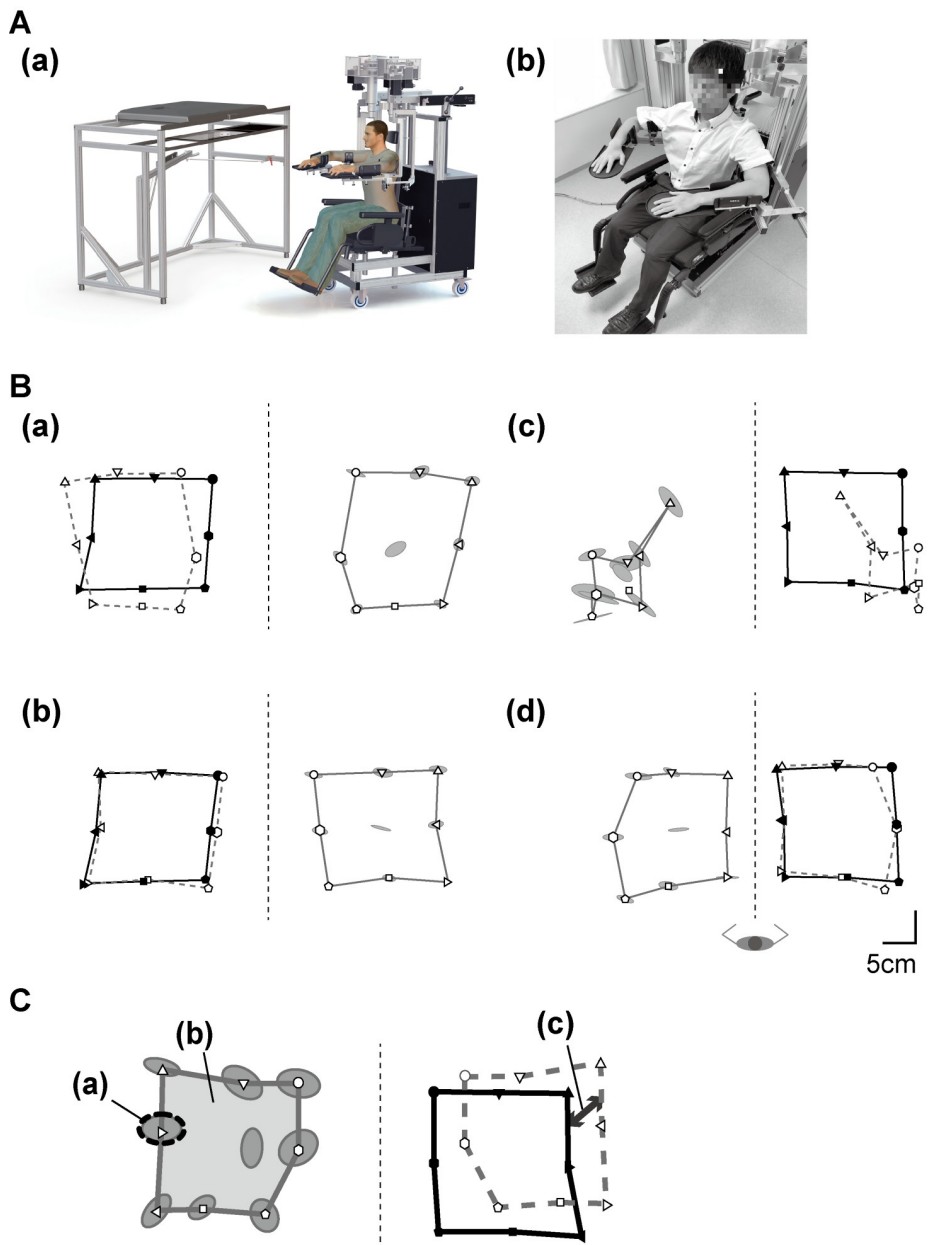

**Fig 2. The KINARM Exoskeleton robotic device and measures of the arm position matching task.** A. Overview of the robotic apparatus. (a) An overall cartoon depiction (courtesy of Kinarm Ltd., Kingston, Ontario). (b) Frontal view of the wheelchair base and exoskeleton arms. In performing the arm position matching task, the central target, which is the starting position, was set such that the shoulder was at approximately 30º of horizontal abduction and the elbow at 90º of flexion. B. Schema of the arm position matching task. Filled symbols (outlined with solid black lines) represent the positions where the robot moves the passive arm. Open symbols (outlined with solid gray lines) represent where the participants matched the perceived mirrored position with the active arm. The performance of the active arm is mirrored on the passive side using gray dashed lines. Ellipses around targets demonstrate variability of six matching attempts and represent one standard deviation. (a) A typical left-hemiparetic patient with the thumb localization test (TLT) score 0 displays little variability and is relatively accurate in matching the arm position with vision occluded. (b) In the same subject as (a), the matching is more accurate with vision restored. (c) A typical right-hemiparetic participant with the TLT score 3 shows increased variability, contraction of the workspace, and poor performance in overall matching with vision occluded. (d) In the same participant as (c), the accuracy of the matching is improved with vision restored. C. Illustration of the parameters. (a) Variability; trial-to-trial variability of the active hand, (b) Area; the overall spatial area matched by the active hand, (c) Shift; systematic shifts between the active and passive hands. The individual pictured in this figure has provided written informed consent (as outlined in PLOS consent form) to publish his image alongside the manuscript.

minutes. The following robotic measures, calculated using Eqs (1) to (3), were used for the analyses (Fig 2C). These measures have been confirmed as having good inter-rater reliability in patients with stroke [20].

- *Variability*: This refers to the trial-to-trial variability of the perceived mirrored location by the active hand. This measure is calculated by calculating the standard deviation of the active hand's position for each target location, then calculating the mean of the standard deviations for all target locations in the x-coordinate (*Variability X*) and the y-coordinate (*Variability Y*). A smaller value for this measure indicates less variability among the repeated attempts.

$$Variability = \sqrt{(Variability\ X)^2 + (Valiability\ Y)^2} \tag{1}$$

- *Area*: This refers to the ratio of the spatial area enclosed by the active hand (representing the perceived workspace of the passive hand) relative to the actual area enclosed by the passive hand. A value of $< 1$ indicates contraction of the perceived area compared to the actual movement area, while a value of $> 1$ indicates expansion of the perceived area. A value of 1 indicates a perfect match.

$$Area = area_{active}/area_{passive} \tag{2}$$

- *Shift*: This indicates the constant bias between the active and passive hands. This measure is calculated by finding the mean error between the active and passive hands for each target location, then calculating the mean of mean errors for all target locations in the x-coordinate (*Shift X*) and the y-coordinate (*Shift Y*). A smaller value indicates less bias in the horizontal plane.

$$Shift = \sqrt{(Shift\ X)^2 + (Shift\ Y)^2} \tag{3}$$

**Analysis.**   The TLT with vision restored was not considered for the analysis as all participants obtained the normal score (0). The TLT score with vision occluded was adopted for further analysis. Quadratically weighted Kappa [26] was used to examine the inter-rater reliability of the TLT. The strength of agreement with kappa statistics were interpreted as follows: $<$ 0.00, poor; 0–0.20, slight; 0.21–0.40, fair; 0.41–0.60, moderate; 0.61–0.80, substantial; and 0.81–1.00, almost perfect agreement [27]. Any disagreements in the scores were discussed by the two raters and rescored. The median value of the scores in the three positions was selected for the following analysis.

Prior to examining the correlation between the TLT and the robotic measures, each robotic measure with vision occluded and vision restored conditions was compared using the Wilcoxon matched-pairs signed-ranks test to ascertain whether or not the values significantly deteriorated without visual cues in the same manner as the TLT. Then, the correlation between each robot-driven measure with vision occluded and the TLT score was investigated using the Spearman's rank correlation coefficients. The 95% confidence interval of the Spearman's rank correlation coefficient was calculated based on Fisher's transformation. The strength of the correlation coefficients was interpreted as described by Guilford [28]: 0.0–0.2, slight; 0.2–0.4, low; 0.4–0.7, moderate; 0.7–0.9, high; and 0.9–1.0, very high correlation. In addition, the values

**Table 1. Participant characteristics.**

| Variables | | (N = 40) |
|---|---|---|
| Age, mean ± SD (range) | | 47.6±12.2 (18–78) |
| Sex, male/female | | 25/15 |
| Type of stroke | Cerebral infarction | 16 |
| | Cerebral hemorrhage | 23 |
| | Subarachnoid hemorrhage | 1 |
| Days after the onset of hemiparesis, median (range) | | 486.5 (164–6456) |
| Part A score of the Fugl-Meyer Assessment of Upper Extremity on the affected side, median (range) | | 22 (5–35) |
| Modified Ashworth scale for elbow flexors in the affected arm, n in [0, 1, 1+, 2, 3, 4] | | [8, 14, 17, 1, 0, 0] |
| Thumb localizing test, n in [0, 1, 2, 3] | | [16, 11, 8, 5] |

of the robotic measure that correlated best with the TLT were compared between the TLT rating groups (0 to 3) using Kruskal-Wallis rank test with Dunn's post-hoc test. Statistical analyses were performed with STATA/SE 13.1 (StataCorp., Texas, USA). Any $P$-value less than 0.05 was considered statistically significant. Through an a priori power analysis for calculating correlation coefficients using G*Power 3.1 [29], we estimated at least a sample size of 29 in total to provide effect size of 0.5, power of 80%, and type 1 error of 0.05.

## Results

The clinical characteristics of the participants are presented in Table 1. Participants with a wide range of motor impairments were included. Spasticity of the participants ranged from mild to moderate. Although all the participants obtained the normal score (0) in the vision restored condition, the TLT determined 24 out of 40 participants (60%) as having proprioceptive impairment with vision occluded condition. The overall weighted kappa of the TLT scores between the two raters was 0.84 ($P < 0.001$), indicating almost perfect agreement (Table 2).

On comparing each of the robotic measures between the two visual conditions, *Variability* and *Shift* were significantly larger with vision occluded than with vision restored (Variability: median value 5.76 cm vs 3.34 cm, $P < 0.001$; Shift: median value 6.35 cm vs 3.81 cm, $P < 0.001$). *Area* was significantly smaller with vision occluded than with vision restored (median value 0.56 vs 0.85, $P = 0.010$) (Figs 3 and 4).

As shown in Table 3, in the correlational analyses between the TLT and the robotic measures, *Area* correlated most strongly with the TLT with vision occluded condition. This correlation was high and negative. The scatterplot of *Area* with the two visual conditions (Fig 4) also implies the decrease of *Area* in the patients with severe impairment (TLT = 2,3). *Variability* showed moderate correlations with the TLT. The correlation between *Shift* and the TLT was not significant.

**Table 2. Scores of the thumb localizing test and quadratically weighted kappa.**

| | Scores of rater A, n in [0, 1, 2, 3] | Scores of rater B, n in [0, 1, 2, 3] | Quadratically weighted kappa | P values |
|---|---|---|---|---|
| Position 1 | [15,17,1,7] | [15,15,5,5] | 0.86 | < 0.001 |
| Position 2 | [15,12,5,8] | [17,7,11,5] | 0.86 | < 0.001 |
| Position 3 | [20,8,9,3] | [17,7,11,5] | 0.81 | < 0.001 |
| Total | [50,37,15,18] | [49,29,27,15] | 0.84 | < 0.001 |

Position 1, the forearm in the neutral position, with the elbow at 90˚ of flexion, and the shoulder at 0˚ of flexion; Position 2, the forearm in the neutral position, with the elbow at 90˚ of flexion, and the shoulder flexed so that the thumb is at the same level as the mouth; Position 3, the forearm in the neutral position, with the elbow at 90˚ of flexion, and the shoulder internally rotated so that the thumb is over the midline of the trunk.

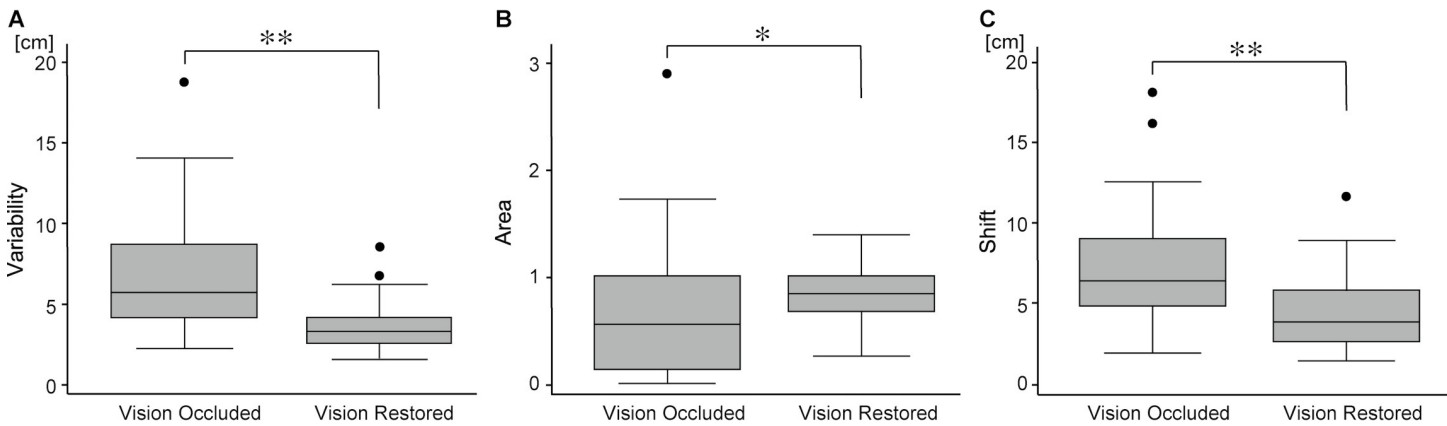

**Fig 3. Comparisons between the measures of arm position matching task with vision occluded and vision restored.** Boxplots of Variability, Area, and Shift are shown with vision occluded (left) and vision restored (right). Error bars indicate standard deviations. ** P < 0.01, * P < 0.05.

On performing the Kruskal-Wallis rank test, the values of *Variability* (P = 0.049) and *Area* (P < 0.001) were found to be significantly different across the TLT rating groups, but the values of *Shift* were not (P = 0.248). Multiple comparisons using Dunn's post-hoc test revealed that the values for *Variability* were significantly different between the TLT scores of 0 and 2 (P = 0.039), 0 and 3 (P<0.001), and 1 and 3 (P = 0.017). As for the values for *Area*, significant differences were seen between the TLT scores of 0 and 1 (P = 0.027), 0 and 2 (P < 0.001), 0 and 3 (P < 0.001), and 1 and 3 (P = 0.015) (Fig 5).

## Discussion

This study examined, for the first time, the reliability of the TLT and its validation against quantitative measure of proprioception derived from the robotic device.

The TLT is the proprioceptive assessment frequently used in both clinical and research settings. Nevertheless, its psychometric properties have not been adequately shown, as was

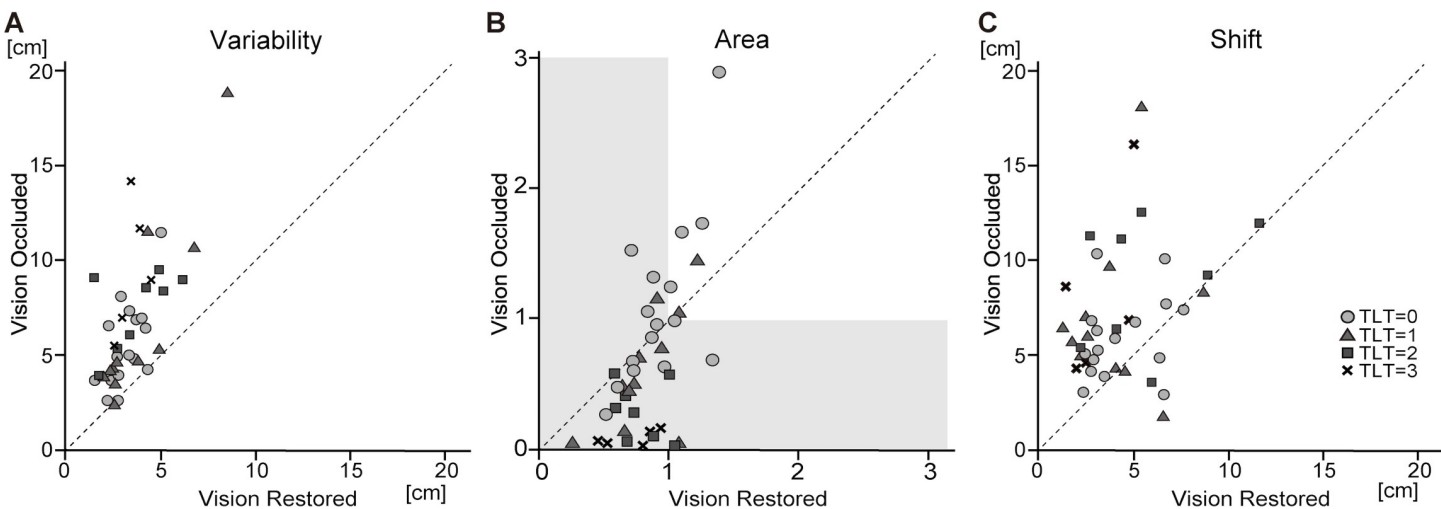

**Fig 4. Scatterplots of robotic measures with vision occluded and with vision restored in each grade of the thumb localizing test (TLT).** Scatterplots of (A)*Variability*, (B)*Area*, and (C)*Shift* are shown. The dots under the dotted line (y = x) indicate that the value is smaller with vision occluded than with vision restored, and the dots above the dotted line (y = x) indicate that the value is larger with vision occluded than with vision restored. The value of Area < 1 (gray shaded area) implies contraction and > 1 implies expansion of the workspace.

**Table 3. Values of the robotic measures with vision occluded and correlations with thumb localizing test.**

| Measures | Median value (range) | Spearman's rho (95% confidence intervals) | P values |
|---|---|---|---|
| *Variability* | 5.76 (2.34–18.78) | 0.40 (0.10–0.63) | 0.011 |
| *Area* | 0.56 (0.01–2.90) | -0.71 (-0.84 - -0.51) | < 0.001 |
| *Shift* | 6.35 (1.81–18.11) | 0.27 (-0.05–0.54) | 0.093 |

discussed in the Background. Contrary to the previous findings indicating the poor inter-rater reliability of some clinical sensory assessments [30], this study revealed that the TLT had almost perfect inter-rater reliability in the standardized way.

All participants in the present study could successfully reach the opposite thumb in the TLT under vision regardless of their TLT scores in the blindfold condition. Similarly, all three robotic measures were significantly worse in the vision occluded condition, compared to the vision restored condition. The improvement in motor performance when aided by visual information in patients with severely impaired proprioception is known [4, 25]. These findings indicate that clinical measures and robotic assessments have the same characteristics. Although some studies report that vision does not always compensate for kinesthetic impairments in stroke [31, 32], the subjects in these studies were in the relatively early phases (average of less than 2 weeks post stroke). On the contrary, the present study was performed in the chronic phase, in which visuospatial deficits or fatigue were more likely to diminish compared to early or subacute phases post stroke, and thus visual compensation was considered to be more effective than the above-mentioned studies.

The TLT scores were significantly correlated with two of three quantitative measures of proprioception derived from the robotic device. Especially, a high correlation was found with *Area* in the vision occluded condition; the values of *Area* decreased as the TLT scores increased. Furthermore, the majority of values for *Area* in the vision occluded condition were < 1 and smaller than those with vision restored, especially in those with TLT ≥ 2. The findings could be interpreted as showing that, without vision, the participants with worse TLT score might perceive the passive limb movement by the robot to be smaller than the actual movement. This result is quite reasonable considering the procedure of the TLT, in which one judges the thumb position after being moved from the preceding position similar to the

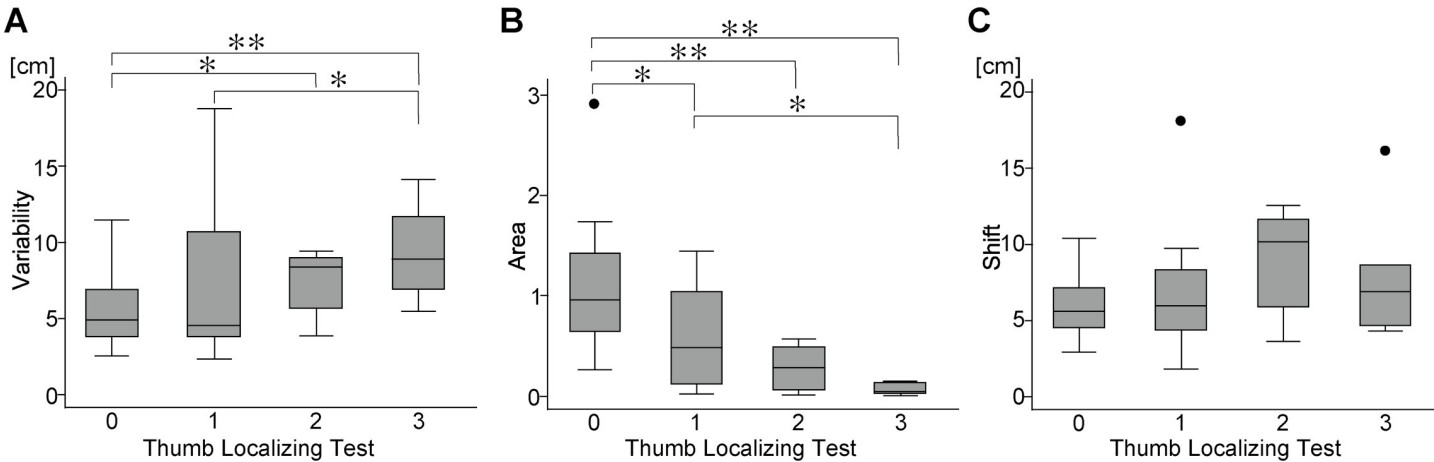

**Fig 5. Robotic measures with vision occluded of the arm position matching task in each grade of the thumb localizing test (TLT).** Boxplots of (A)*Variability*, (B)*Area*, and (C)*Shift* are shown. Error bars indicate standard deviations and the lines within the box indicate median values. ** P < 0.01, * P < 0.05 with the Dunn's test for multiple comparisons.

robotic task, and it also indicates that the TLT may reflect at least a part of "kinesthetic sensation", which is generated from muscle spindles in response to limb movement and changes in position [33]. The TLT scores also correlated moderately with *Variability*. The result seems to be reasonable because *Variability* is similar to the TLT rating in that it evaluates the gap or sway of the active limb. This measure was also previously reported as having a significant association with the TLT in its ability to discriminate between the normal and abnormal [22, 34], which was confirmed by our study. On the other hand, this study did not show significant correlation between the TLT and *Shift*. The measure reflects the bias of the expression of the perceived area, and could be influenced by rather other factors such as spatial cognition than proprioception. Therefore, no correlation with *Shift* may not hamper the validity of the TLT for proprioceptive sensation.

Although no study has examined the correlation between TLT and quantitative measures, i.e. Area, Variability, and Shift, some conflicts were found in a few previous studies [20, 22], where no categorical relationships were demonstrated between the TLT and *Area* in stroke patients, and significant categorical relationships were found between the TLT and *Shift*. This difference in the findings may be related to two reasons. First, in our study, an almost perfect inter-rater reliability in the TLT was achieved by standardizing the details of testing and the rating. This is the strong point of the present study, ensuring a robustness of the following analysis. Second, our study investigated chronic patients living independently, which suggests they have mild, if any, cognitive problems. In contrast, the previous studies [20, 22] investigated subacute patients whose functional abilities are various, and thus might have various confounding factors such as disturbance in spatial cognition due to cognitive impairments, including visuospatial neglect [31] or deficits in generalized attention. Our study might successfully focus on sensorimotor impairments with less confounding factors.

In the comparisons between the grades of TLT, *Variability* and *Area* were not statistically different and substantially overlapped between grade 1 and 2, and between 2 and 3 of the TLT score. This result suggests that the TLT rating may have some ambiguity in assessing the degree of proprioceptive deficits between these grades. Clinicians should bear in mind that the difference of 1 point in the TLT score between grade 1–3 should not be emphasized. In the context of robotic devices being more objective and quantitative than clinical assessment [35–37], the robotic assessment can be considered for further assessment of quantification of proprioception such as research purpose.

This study had a few limitations. First, another study design is needed to study test-retest reliability of the TLT. Second, the generalizability of our findings may be somewhat limited as the present study investigated chronic patients living independently, who were relatively young and high functioning compared to the general stroke population. In addition, further investigation will be needed to generalize our findings to individuals with other pathological conditions. Finally, another study would be required to reveal the relationship between the proprioceptive deficits represented by the TLT and brain lesions that may affect passive kinesthesic sensation or sensorimotor integration, as indicated in recent studies [8, 31].

## Conclusions

The present study demonstrated that the TLT had a high inter-rater reliability when the testing procedure was standardized. The TLT correlated highly with the quantitative measures which reflects the perceived area of movement and variability of the limb location. These findings contribute to the robustness of the TLT as a clinical measure. Furthermore, the overlapping of values in quantitative measures between consecutive grades 1–3 of the TLT gave valuable information for interpretation of the TLT score in the clinical settings. In spite of the ordinal

nature of the scale, which tends to be a disadvantage when compared with continuous scales, the TLT has proven to be essential as an easily performed and robust tool to detect proprioceptive deficits in clinical scenes.

## Supporting information

**S1 Data. Data for assessing the inter-rater reliability of the TLT.** Details of the variables are as follows: SubjectNo, an anonymous subject number for the present study only; PareticSide_R1L2, paretic side (1 = right, 2 = left); TL_sv_A1/2/3, the scores of TLT with vision occluded condition by one of the two raters; TL_sv_B1/2/3, the scores of TLT with vision occluded condition by another rater; each number describes three limb positions, as defined in the manuscript (1 = Fig 1C(a), 2 = Fig 1C(b), 3 = Fig 1C(c)).
(XLSX)

**S2 Data. Data of participant characteristics, clinical measures, and robotic measures.** Details of the variables are as follows: TYPE_C1_H2_S3, the type of diseases (1 = cerebral infarction, 2 = cerebral hemorrhage, and 3 = subarachnoid hemorrhage); PareticSide_R1L2, paretic side (1 = right, 2 = left); MAS_elbow/wrist/finger, Modified Ashworth scale, originally scaled as 0, 1, 1+, 2, 3, 4, described in the dataset as 0, 1, 2, 3, 4, 5 respectively; Variability_sv, Area_sv, Shift_sv, robotic measures obtained with vision occluded condition; Variability_cv, Area_cv, Shift_cv, robotic measures with vision restored condition; TL_sv_1, TL_sv_2, TL_sv_3, the ultimate TLT scores discussed and rescored by the two raters; TL_sv_median, the median value of the TLT scores in the three limb positions.
(XLSX)

## Acknowledgments

We thank Ms. Sawako Ohtaki and Ms. Miho Hiramoto for their technical support.

## Author Contributions

**Conceptualization:** Eri Otaka, Yohei Otaka.

**Data curation:** Eri Otaka, Yohei Otaka, Shoko Kasuga, Atsuko Nishimoto, Kotaro Yamazaki.

**Formal analysis:** Eri Otaka.

**Methodology:** Eri Otaka, Yohei Otaka.

**Supervision:** Michiyuki Kawakami, Junichi Ushiba, Meigen Liu.

**Writing – original draft:** Eri Otaka.

**Writing – review & editing:** Eri Otaka, Yohei Otaka, Shoko Kasuga, Atsuko Nishimoto, Kotaro Yamazaki, Michiyuki Kawakami, Junichi Ushiba, Meigen Liu.

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
