## [Decision Letter · Decision Letter 0]

24 Mar 2020

PONE-D-20-06716

Reliability of Thumb Localizing Test and its validity against quantitative measures with a robotic device in patients with hemiparetic stroke

PLOS ONE

Dear Dr. Otaka,

Thank you for submitting your manuscript to PLOS ONE. After careful consideration, we feel that it has merit but does not fully meet PLOS ONE’s publication criteria as it currently stands. Therefore, we invite you to submit a revised version of the manuscript that addresses the points raised during the review process.

As you will see below, the two Reviewers point to some issues regarding the methodological approach and the interpretation of the results.  Reviewer#1 asks for clarifications about the methods and more detailed descriptions of the tests used (e.g. Fugl-Meyer). The Reviewer also has several suggestions to improve the Introduction and Discussion. Reviewer#2 has some concerns regarding the TLT scoring scheme and how this might have affected the reliability. Please make sure that all concerns are adequately addressed in the revised version.

We would appreciate receiving your revised manuscript by May 08 2020 11:59PM. To enhance the reproducibility of your results, we recommend that if applicable you deposit your laboratory protocols in protocols.io, where a protocol can be assigned its own identifier (DOI) such that it can be cited independently in the future. For instructions see: http://journals.plos.org/plosone/s/submission-guidelines#loc-laboratory-protocols

We look forward to receiving your revised manuscript.

Kind regards,

François Tremblay, PhD

Academic Editor

PLOS ONE

Journal Requirements:

"M.L., M.K. and J.U. are the founding scientists of Connect Inc., a commercial company for the development of rehabilitation devices since May 2018. They have received a salary from Connect Inc., and have held the shares in Connect Inc. They hold the managerial positions at Connect Inc. Their conditions were disclosed to the Universities, and were approved. Connect Inc. does not have any relationship with the present study. The other authors report no conflict of interest."

3. We note that Figure 1 includes an image of a participant in the study.

Reviewers' comments:

Reviewer's Responses to Questions

**Comments to the Author**

1. Is the manuscript technically sound, and do the data support the conclusions?

Reviewer #1: Yes

Reviewer #2: Yes

2. Has the statistical analysis been performed appropriately and rigorously? 

Reviewer #1: Yes

Reviewer #2: Yes

3. Have the authors made all data underlying the findings in their manuscript fully available?

Reviewer #1: Yes

Reviewer #2: Yes

4. Is the manuscript presented in an intelligible fashion and written in standard English?

Reviewer #1: Yes

Reviewer #2: Yes

5. Review Comments to the Author

Reviewer #1: Dear study authors

Thank you for the opportunity to review your manuscript. You have undertaken a study to determine the inter-rater validity of the thumb localising test in a chronic stroke population, and to examine its correlation with the KINARM exoskeleton device.

Firstly, thank you for the figures. They are very useful in these types of studies. Your study is nicely summarised and clear, however I feel you are currently missing a few important details that would assist readers. I have provided some comments below for your consideration. I hope these are helpful.

Major comments.

Introduction / References: Your background studies and references and all starting to get quite old. There is newer work in this area. Can you please incorporate discussion of more recent findings throughout the manuscript, and further discuss the implications for your methods and results. How does this work fit into other studies in both stroke and proprioception?

Methods: As you are undertaking an inter-rater validity study, please provide details on the raters, including their training, experience, and professional background). Was the person helping the participant a different person to the assessors, and if so please also add details on who they are.

Please explain the Fugl-Meyer and the Ashworth tests in more detail, including how you undertook them, the scoring (max score and how scored) and interpretation of the scores (i.e. a high Fugl-Meyer score is?). Also, please state which side you are reporting the Fugle-Meyer for (ie. More affected, less affected).

What are the time intervals between testing, were they all done on the same day, different days etc. how did you manage fatigue in these participants?

Results: Please put the results for each variable in a table in addition to the figures. Especially for the TLT results which do not appear to be reported at present.

Discussion: There are some terms in your discussion that have not been introduced earlier. Please give a short description of what you mean by these terms (i.e. kinaesthetic sensation, spatial cognition. How do you think the changes that may have occurred in the brain after the stroke has influenced your findings?

Limitations: You could expand on your limitations section, particularly;

You have briefly mentioned it, but you seen to have a fairly young and very high functioning cohort. This does not seem overly typical of a normal stroke population. Similarly, do you think your test would be as accurate in a less high functioning population? Please expand on this limitation.

Are you likely to see these changes with age, or are they purely due to the stroke?

You have mentioned in the introduction the TLT is only an ordinal test, however what does this for interpretation of your results.

Minor comments:

Table 1: You may wish to relabel “types of disease”, as Stroke type, or type of stroke.

Line 330 – typo – “grades” rather than grads.

309 – do you mean the study did “not” show significant correlations?

Reviewer #2: This is a well-written manuscript examining the inter-rater reliability of the Thumb Localizer Task in 40 participants with chronic stroke. Additionally, the authors make comparisons of the TLT results with those obtained from a well validated robotic measure of proprioception called position matching performed in the KINARM robot. The methodologic approach appears quite reasonable. My main suggestions with the paper are fairly minor.

The authors might consider making the point that the TLT is only a 4 point scale, and although it would theoretically be possible to get low reliability when doing such an experiment, not obtaining high inter-rater reliability from two trained observers scoring the same video of a patient would present quite a challenge.

The authors should also take a little more time to explain why they used the “vision restored” conditions.

“though it is a four-point scale and thus unable to quantify the deficits.” Although I agree with the authors a 4 point ordinal scale isn’t exactly the most desirable tool to “quantify” something. However the definition of the word “quantify” basically means you are trying to measure something and the TLT does this. I would suggest rewording the sentence accordingly.

Figures 1, 2 – The image on the PDF file I received was quite pixelated. The authors should consider addressing this prior to publication (I have seen this happen with some of my own manuscripts when the file gets converted by the publisher’s software, sometimes changing the file type can be helpful).

Discussion

Figures 4, 5 - It would have been nice here to also see Variability and Shift plotted in a similar way against TLT. Consider turning these into 3 panel figures to display all the data.

Page 14 – “significantly deteriorated” implies that the vision restored condition was conducted first, then the vision occluded. I suspect the authors meant “was significantly worse” or similar.

The authors may want to consider reviewing/discussing/contrasting a larger study investigating the return of vision (Herter et al. Vision does not always help stroke survivors compensate for impaired limb position sense. J Neuroeng Rehabil 2019.) as it uses the same task and many subjects fail to improve with vision restored.

6. PLOS authors have the option to publish the peer review history of their article (what does this mean?). If published, this will include your full peer review and any attached files.

Reviewer #1: No

Reviewer #2: No

---

## [Author Response · Author response to Decision Letter 0]

22 May 2020

Please see the uploaded file named Respond to Reviewers.

---

## [Decision Letter · Decision Letter 1]

23 Jun 2020

PONE-D-20-06716R1

Reliability of the Thumb Localizing Test and its validity against quantitative measures with a robotic device in patients with hemiparetic stroke

PLOS ONE

Dear Dr. Otaka,

Thank you for submitting your manuscript to PLOS ONE. After careful consideration, we feel that it has merit but does not fully meet PLOS ONE’s publication criteria as it currently stands. Therefore, we invite you to submit a revised version of the manuscript that addresses the points raised during the review process.

As you will see below, the Reviewers were mostly satisfied with the latest version. Reviewer #2 has some minor points that will require your attention. I urge you to proceed with these corrections with diligence, as there will be no need for another round of review, provided that the minor points are adequately addressed.

We look forward to receiving your revised manuscript.

Kind regards,

François Tremblay, PhD

Academic Editor

PLOS ONE

Reviewers' comments:

Reviewer's Responses to Questions

**Comments to the Author**

1. If the authors have adequately addressed your comments raised in a previous round of review and you feel that this manuscript is now acceptable for publication, you may indicate that here to bypass the “Comments to the Author” section, enter your conflict of interest statement in the “Confidential to Editor” section, and submit your "Accept" recommendation.

Reviewer #1: All comments have been addressed

Reviewer #2: (No Response)

2. Is the manuscript technically sound, and do the data support the conclusions?

Reviewer #1: (No Response)

Reviewer #2: Yes

3. Has the statistical analysis been performed appropriately and rigorously? 

Reviewer #1: (No Response)

Reviewer #2: Yes

4. Have the authors made all data underlying the findings in their manuscript fully available?

Reviewer #1: (No Response)

Reviewer #2: Yes

5. Is the manuscript presented in an intelligible fashion and written in standard English?

Reviewer #1: (No Response)

Reviewer #2: Yes

6. Review Comments to the Author

Reviewer #1: (No Response)

Reviewer #2: Thanks to the authors for responding to my previous review. The paper appears much improved. I did catch a few minor details that they authors should likely deal with.

Line 50 – actually the longitudinal process following stroke recovery has been discussed in a few papers, one that uses the same device the authors are using in the present manuscript (see Semrau et al. 2015, Stroke “Examining Differences in Patterns of Sensory and Motor Recovery After Stroke With Robotics”. There are also a few papers that use clinical scales to look at recovery of proprioception and a reasonably sized recent one published in Neurorehabilitation and Neural Repair (“Zandvliet et al. 2020 34(5):403-426 “Is Recovery of Somatosensory Impairment Condition for Upper-limb motor recovery after stroke?”). So the authors may want to soften the existing comment and reference studies that have looked at it. I completely agree there is room for further, more detailed studies here, but they authors should not ignore the efforts of others that have already looked at this issue.

Line 106 – Fugl-Meyer. The upper score should be 66 here (I think the 36 is a typo).

Line 314 – I suspect you mean “quantitative” rather than “qualitative” measure as it is written.

7. PLOS authors have the option to publish the peer review history of their article (what does this mean?). If published, this will include your full peer review and any attached files.

Reviewer #1: No

Reviewer #2: No

---

## [Author Response · Author response to Decision Letter 1]

1 Jul 2020

Please see the Response to Reviewers.

---

## [Editor Report · Decision Letter 2]

8 Jul 2020

Reliability of the Thumb Localizing Test and its validity against quantitative measures with a robotic device in patients with hemiparetic stroke

PONE-D-20-06716R2

Dear Dr. Otaka,

We’re pleased to inform you that your manuscript has been judged scientifically suitable for publication and will be formally accepted for publication once it meets all outstanding technical requirements.

Kind regards,

François Tremblay, PhD

Academic Editor

PLOS ONE
---

## [Editor Report · Acceptance letter]

13 Jul 2020

PONE-D-20-06716R2 

Reliability of the Thumb Localizing Test and its validity against quantitative measures with a robotic device in patients with hemiparetic stroke 

Dear Dr. Otaka:

I'm pleased to inform you that your manuscript has been deemed suitable for publication in PLOS ONE. Congratulations! Your manuscript is now with our production department. 

Kind regards, 

on behalf of

Dr. François Tremblay 

Academic Editor

PLOS ONE